# DNA Methylation Inhibition Reversibly Impairs the Long-Term Context Memory Maintenance in *Helix*

**DOI:** 10.3390/ijms241814068

**Published:** 2023-09-14

**Authors:** Alena B. Zuzina, Aliya Kh. Vinarskaya, Pavel M. Balaban

**Affiliations:** Cellular Neurobiology of Learning Lab, Institute of Higher Nervous Activity and Neurophysiology, Russian Academy of Sciences, 5A Butlerova St., Moscow 117485, Russia; lucky-a89@mail.ru (A.B.Z.); aliusha1976@mail.ru (A.K.V.)

**Keywords:** epigenetics, DNA methylation, RG108, context memory, memory maintenance, gastropods, reconsolidation

## Abstract

This work aims to study the epigenetic mechanisms of regulating long-term context memory in the gastropod mollusk: *Helix*. We have shown that RG108, an inhibitor of DNA methyltransferase (DNMT), impaired long-term context memory in snails, and this impairment can be reversed within a limited time window: no more than 48 h. Research on the mechanisms through which the long-term context memory impaired by DNMT inhibition could be reinstated demonstrated that this effect depends on several biochemical mechanisms: nitric oxide synthesis, protein synthesis, and activity of the serotonergic system. Memory recovery did not occur if at least one of these mechanisms was impaired. The need for the joint synergic activity of several biochemical systems for a successful memory rescue confirms the assumption that the memory recovery process depends on the process of active reconsolidation, and is not simply a passive weakening of the effect of RG108 over time. Finally, we showed that the reactivation of the impaired memory by RG108, followed by administration of histone deacetylase inhibitor sodium butyrate, led to memory recovery only within a narrow time window: no more than 48 h after memory disruption.

## 1. Introduction

DNA methyltransferase (DNMT) is an enzyme that modifies cytosine in DNA to 5-methylcytosine [1,2]. It has been consistently shown that DNMT negatively regulates the transcription process [3,4,5,6,7,8], as originally hypothesized more than 20 years ago by Holliday [9]. A number of studies demonstrated that the DNMT-mediated increase in methylation and a resulting decreased gene expression affects memory consolidation [1,3,4,5,9,10,11,12,13,14,15,16,17,18,19,20,21,22,23,24,25], as well as memory reconsolidation [26,27,28,29,30,31]. Disrupting DNA methylation through treatment with inhibitors of DNMT blocks the induction and maintenance of memory as a part of epigenetic regulation of memory formation [32]. It should be noted that in contrast to studies in mammals, very little is known about the role of DNA methylation in memory processes in mollusks. DNA methylation effects were demonstrated in gastropods using a long-term facilitation paradigm [33], intermediate-term learning [34], long-term memory (LTM) in *Aplysia* [35,36], and operantly conditioned respiratory behavior in *Lymnaea* [37,38,39].

Another important epigenetic regulatory mechanism is the histone acetylation. In contrast to DNA methylation, an increase in histone acetylation causes changes in chromatin structure and promotes enhanced transcription in most cases [40,41,42,43]. Histone acetylation, as well as DNA methylation, is widely involved in the regulation of synaptic plasticity, memory consolidation, and reconsolidation processes [5,10,14,16,17,20,44,45,46,47,48,49,50,51]. The fact that DNA methylation and histone acetylation can influence each other has been shown. Wade with colleagues demonstrated that DNA methylation recruits complexes that include histone deacetylases (HDACs), which lead to removing acetyl groups from histones [52]. On the contrary, it was shown that histone acetyltransferases (HATs) somehow affect an active DNA demethylation in plants [53]. Another study has shown that DNMT inhibitors reduced HAT expression and led to decrease in H3 and H4 acetylation [54]. Recent studies showed that the suppression of DNMT activity led to a decrease in histone acetylation and memory deficit [5,20,26,55] whereas pretreatment with an HDAC inhibitor prevented the DNMT inhibitor-induced memory deficit [5,20].

It is clear that DNA methylation regulates a wide range of learning and memory tasks among different species. However, the involvement of DNMT in context memory maintenance and reconsolidation in gastropod mollusk *Helix* is not known yet. In the current work, it was shown for the first time that the reconsolidation can develop normally under the DNMT inhibition: the inhibition of DNA methylation by RG108 in combination with memory reactivation did not cause context memory impairment in *Helix*. RG108 administration that occurred without reminding, impaired maintenance of long-term context memory in *Helix* and, surprisingly, this effect was reversible. It appeared that memory reinstatement depended on the simultaneous coordinated work of several systems: nitric oxide (NO) synthesis, protein synthesis, activity of the serotonergic system, and did not occur if at least one of these mechanisms was impaired. To discover whether the context memory impairment under the DNMT inhibition can be rescued, we used such cognitive enhancer as sodium butyrate (NaB)—HDAC inhibitor. We showed that memory reactivation followed by administration of NaB led to memory recovery but only within a narrow time window (no more than 48 h after memory disruption).

## 2. Results

### 2.1. DNMT Inhibition Impairs Context Memory Maintenance but Not Reconsolidation

The aim of the first experiment was to investigate the role of DNMT activity in the maintenance of long-term context memory in snails and reconsolidation of this memory following retrieval. We examined the effects of the DNA methyltransferase inhibitor RG108 on these processes. Snails were divided in three groups RG48-72 (*n* = 11), RG48-72 + R (*n* = 11), and Veh (*n* = 11) (Figure 1A–C). Snails were given context training with shocks in context 1 only for 10 days and tested for retention of context memory 24 h later (test T1). ANOVA revealed a reliable main effect of context F(1,60) = 551.8, *p* < 0.001: withdrawal responses in two contexts in all groups were significant differed. The next day the group RG48-72 was injected with RG108 without reminder, RG48-72 + R was injected with RG108 one hour following the memory reactivation (R = reminder = 20 min on a ball), and control group Veh was vehicle-injected. A total of 24 h later (test T2) ANOVA revealed a reliable main effect of context F(1,60) = 675.7, *p* < 0.001; group F(2,60) = 24.5, *p* < 0.001; and a reliable interaction F(2,60) = 27.4, *p* < 0.001. Post hoc analysis revealed that RG48-72 exhibited significantly impaired LTM (9.1 ± 1.5%) relative to the Veh group (75.8 ± 5.7%); RG48-72 + R (68.5 ± 5.3%) and Veh groups exhibited similar high levels of withdrawal responses on the ball during test T2. The following day all groups were tested for LTM once again (test T3): ANOVA revealed a reliable main effect of context F(1,60) = 527.9, *p* < 0.001; group F(2,60) = 37.0, *p* < 0.001; and a reliable interaction F(2,60) = 39.5, *p* < 0.001. Post hoc analysis revealed that RG48-72 (52.9 ± 5.1%), RG48-72 + R (76.6 ± 6.9%), and Veh group (67.7 ± 3.9%) did not differ significantly in the conditioned reactions on the ball. Next-day testing (T4) revealed that all groups (RG48-72—58.9 ± 3.4%, RG48-72 + R—66.6 ± 7.8%, and Veh—74.2 ± 3.4%) demonstrated good context memory and did not differ significantly. Thus, we observed that inhibition of DNMT activity by RG108 impairs the long-term context memory (Figure 1A, T2). Then, we showed that fear memory that should be disrupted by RG108 makes full recovery when animals are exposed to the reactivation of memory after RG108 injection (Figure 1B). Obtained results can be interpreted as a rescue effect of memory reactivation on the long-term context memory.

In the second series of experiments, we tested whether the memory impaired by RG108 would be restored if memory would not be reactivated in the first 24 h or 48 h after administration of the DNMT inhibitor (i.e., test sessions T2 was performed 48 h after T1 (as described above in Figure 1) but test session T3 was made 96 h (not 72 h as described above) after T1). The protocol of the experiment was similar to the previous one, except we retested the maintenance of LTM in 24 h (T2) and 72 h (T3) after the RG108 treatment. Snails were divided in two groups RG48-96 (*n* = 8) and Veh (*n* = 11) (Figure 2A). After training, ANOVA revealed a reliable main effect of context F(1,34) = 192.7, *p* < 0.001. Post hoc comparisons revealed that both groups showed highly significant difference between responses in two contexts. Next day the RG48-96 group was injected with RG108 without a reminder. Group Veh served as a control and was injected with vehicle without the reminder. The following day both groups were tested for LTM (test T2): ANOVA revealed a reliable main effect of context F(1,34) = 252.2, *p* < 0.001; group F(1,34) = 38.6, *p* < 0.001; and a reliable interaction F(1,34) = 39.7, *p* < 0.001. Post hoc analysis revealed that RG48-96 exhibited an impaired LTM (9.8 ± 1.1%) relative to the Veh group (75.8 ± 5.7%) at test T2. After 48 h of rest both groups were tested for LTM once again (test T3): ANOVA revealed a reliable main effect of context F(1,34) = 251.3, *p* < 0.001; group F(1,34) = 178.3, *p* < 0.001; and a reliable interaction F(1,340) = 169.4, *p* < 0.001. Post hoc analysis revealed that RG48-96 (14.4 ± 2.9%) showed significantly lower withdrawal reactions on the ball than the control Veh group (68.6 ± 3.5%) that exhibited good memory throughout all the test sessions.

In the next experiment we retested LTM in 24 h (T2) and 96 h (T3) after the RG108 treatment. Snails were divided into two groups RG48-120 (*n* = 7) and Veh (*n* = 11) (Figure 2B). After training, ANOVA revealed a reliable main effect of context F(1,32) = 214.9, *p* < 0.001. Post hoc comparisons revealed that all groups showed highly significant difference between responses in two contexts. The next day the RG48-120 was injected with RG108 without a reminder. Group Veh served as a control and was injected with vehicle without a reminder. The following day both groups were tested for LTM (test T2): ANOVA revealed a reliable main effect of context F(1,32) = 255.5, *p* < 0.001; group F(1,32) = 32.1, *p* < 0.001; and a reliable interaction F(1,32) = 37.5, *p* < 0.001. Post hoc analysis revealed that RG48-120 exhibited impaired LTM (12.0 ± 1.3%) relative to the Veh group (75.8 ± 5.7%). A total of 72 h later both groups were tested for LTM once again (test T3): ANOVA revealed a reliable main effect of context F(1,32) = 219.0, *p* < 0.001; group F(1,32) = 152.5, *p* < 0.001; and a reliable interaction F(1,32) = 164.7, *p* < 0.001. Post hoc analysis revealed that RG48-120 (12.4 ± 2.0%) showed significantly lower withdrawal reactions on the ball than the control Veh group (63.1 ± 3.7%).

In the last experiment in this series, the first testing after RG108 administration was carried out 48 h later (T2); the next testing was carried out in 24 h after T2 (T3). Snails were divided in two groups RG72-96 (*n* = 11) and Veh (*n* = 11) (Figure 2C). After the conditioning procedure ANOVA revealed a reliable main effect of context F(1,40) = 275.5, *p* < 0.001. Post hoc comparisons revealed that all groups showed highly significant difference between responses in two contexts. The next day RG72-96 was injected with RG108 without reminder. Group Veh served as a control and was injected with vehicle. A total of 48 h later both groups were tested for LTM (test T2): ANOVA revealed a reliable main effect of context F(1,40) = 339.4, *p* < 0.001; group F(1,40) = 32.6, *p* < 0.001; and a reliable interaction F(1,40) = 46.4, *p* < 0.001. Post hoc analysis revealed that RG72-96 did not exhibit LTM (7.3 ± 0.7%) relative to Veh group (67.7 ± 3.9%) during test T2. The following day both groups were tested for LTM once again (test T3): ANOVA revealed a reliable main effect of context F(1,40) = 298.0, *p* < 0.001; group F(1,40) = 276.1, *p* < 0.001; and a reliable interaction F(1,40) = 305.6, *p* < 0.001. Post hoc analysis revealed that RG72-96 (8.0 ± 1.6%) and Veh group (68.6 ± 3.5%) were significantly different in the conditioned reaction on the ball.

### 2.2. The Rescue Effect of Context Memory Reactivation Depends on Nitric Oxide Synthesis, Protein Synthesis, and the Activity of the Serotonergic System

In the next series of experiments, we decided to investigate whether the reinstatement of memory impaired after the DNMT inhibition (Figure 1A) is dependent on serotonin, NO production, protein synthesis. Snails were divided in 5 groups: RG48-72/L-NAME (*n* = 13), RG48-72/MET (*n* = 11), RG48-72/ANI (*n* = 12), RG48-72/Veh (*n* = 16), and Veh/Veh (*n* = 12) (Figure 3A–F). After the training procedure, ANOVA revealed a reliable main effect of context F(1,94) = 860.2, *p* < 0.001 (Figure 3B–F, T1). Post hoc comparisons revealed that all groups showed highly significant difference between responses in two contexts. Next-day groups RG48-72/L-NAME, RG48-72/MET, RG48-72/ANI, and RG48-72/Veh were injected with RG108 without reminding. Group Veh/Veh served as a control and was injected with vehicle. The following day all groups were retested for LTM: ANOVA revealed a reliable main effect of context F(1,94) = 987.5, *p* < 0.001; group F(3,94) = 26.3, *p* < 0.001; and a reliable interaction F(3,94) = 29.7, *p* < 0.001 (Figure 3B–E, T2). Post hoc analysis revealed that there was no difference between responses in two contexts in groups RG48-72/L-NAME, RG48-72/MET, RG48-72/ANI, and RG48-72/Veh. These groups exhibited impaired LTM (RG48-72/L-NAME—6.3 ± 1.3%, RG48-72/MET—7.4 ± 0.9%, RG48-72/ANI—6.0 ± 0.7%, RG48-72/Veh—9.9 ± 1.3%) relative to the Veh/Veh group (61.1 ± 3.2%). Immediately after the testing session T2 RG48-72/L-NAME received a L-NAME injection, RG48-72/MET was given a MET injection, RG48-72/ANI received an injection of ANI, and RG48-72/Veh received a sham injection of a saline. Control groups Veh/Veh received a sham injection of a saline at the same time point as all other groups. Testing a day after the drug injections revealed a reliable main effect of context F(1,94) = 303.5, *p* < 0.001; group F(3,94) = 94.7, *p* < 0.001; and a reliable interaction F(3,94) = 114.1, *p* < 0.001 (Figure 3B–E, T3). LTM in group RG48-72/Veh was completely reinstated (54.2 ± 3.8%), while the groups RG48-72/L-NAME (7.9 ± 1.4%), RG48-72/MET (10.1 ± 1.3%), and RG48-72/ANI (6.6 ± 1.6%) still demonstrated no difference between responses in two contexts. Post hoc analysis revealed that the withdrawal response on the ball in a group RG48-72/Veh was not different from that of Veh/Veh (53.2 ± 5.5%) and the withdrawal responses on the ball in RG48-72/L-NAME, RG48-72/MET, and RG48-72/ANI were significantly different from those of RG48-72/Veh and Veh/Veh. In summary, reinstatement of the long-term context memory impaired under the DNMT inhibition depends on NO/ protein synthesis and serotonergic system activation during reminding because reinstatement was not observed under NO/ protein synthesis inhibition and serotonergic receptors blockade.

### 2.3. Inhibition of HDAC Activity Is Able to Rescue Context Memory Deficit Induced by a DNMT Inhibitor Only over a Limited Time Period

The next series of experiments consisted of three separate parts, but all of them were aimed to discover whether context memory impaired under DNMT inhibition can be facilitated under NaB (epigenetic regulator) administration. In the first part of experiments NaB was applied in 24 h after memory impairment and LTM was retested after that in 24 h (Figure 4A–D). Snails were divided in three groups RG48-72/NaB (*n* = 8), RG48-72/Veh (*n* = 8), and Veh/Veh (*n* = 6). They were given shocks in one context only (on the ball), and the percentage of maximal tentacle withdrawal was tested 24 h after training. All groups demonstrated highly significant difference between responses in two contexts (Figure 4B–D, T1, a reliable main effect of context F(1,38) = 302.6, *p* < 0.001). The next day after test T1 the trained animals from RG48-72/NaB and RG48-72/Veh received an intrahemocoelic injection of RG108 without reminding. Veh/Veh was treated with vehicle. RG108 had significantly reduced levels of withdrawal response on ball in RG48-72/NaB (11.4 ± 1.7%) and RG48-72/Veh (9.2 ± 1.5%) compared with Veh/Veh (67.3 ± 8.7%) (Figure 4B–D, T2, a reliable main effect of context F(1,38) = 325.3, *p* < 0.001; group F(2,38) = 12.8, *p* < 0.001; and a reliable interaction F(2,38) = 12.7, *p* < 0.001). Immediately after testing session T2 we administered injections of NaB for group RG48-72/NaB and injections of vehicle for groups RG48-72/Veh and Veh/Veh. A total of 24 h later ANOVA revealed a reliable main effect of context F(1,38) = 259.3, *p* < 0.001; group F(2,38) = 15.4, *p* < 0.001; and a reliable interaction F(2,38) = 12.0, *p* < 0.001 (Figure 4B–D, T3). Post hoc analysis revealed that the withdrawal responses on the ball in RG48-72/NaB and RG48-72/Veh at T3 were significantly different from those at T2. In addition, we found that RG48-72/Veh (54.0 ± 4.7%) and RG48-72/NaB (70.7 ± 7.1%) showed a significant difference in withdrawal responses to tactile stimulation on the ball at T3. This experiment demonstrated that the HDAC inhibitor NaB was capable of improving the RG108-impaired context memory if it is applied in 24 h after memory impairment followed by a repeated memory testing.

The design of the next part of the experiments was similar to the previous one, except we retested LTM in 48 h after NaB treatment. We trained three groups of animals: RG48-96/NaB (*n* = 8), RG48-96/Veh (*n* = 7), and Veh/Veh (*n* = 10). Figure 4E–H shows that training led to a significant increase in withdrawal reaction on the ball relative to the glass in all groups (Figure 4F–H, T1, a reliable main effect of context F(1,60) = 553.8, *p* < 0.001). A total of 24 h later RG48-96/NaB and RG48-96/Veh were injected with RG108 without reminding, whereas Veh/Veh group received vehicle injection. The RG108 group had significantly reduced levels of withdrawal response: RG48-96/NaB (10.7 ± 1.7%) and RG48-96/Veh (7.9 ± 1.5%) compared with Veh/Veh (61.8 ± 3.9%) (Figure 4F–H, T2, a reliable main effect of context F(1,44) = 787.0, *p* < 0.001; group F(2,44) = 34.1, *p* < 0.001; and a reliable interaction F(2,44) = 35.9, *p* < 0.001). Immediately after testing session T2 we administered injections of NaB for group RG48-96/NaB, injections of vehicle for groups RG48-96/Veh and Veh/Veh. A total of 48 h later ANOVA revealed a reliable main effect of group F(2,44) = 102.0, *p* < 0.001; context F(1,44) = 174.8, *p* < 0.001; and a reliable interaction F(2,44) = 102.1, *p* < 0.001 (Figure 4F–H, T3). Post hoc analysis revealed that the withdrawal responses on ball in RG48-96/NaB (16.5 ± 3.9%) and RG48-96/Veh (9.3 ± 1.0%) at test T3 were significantly different from those in Veh/Veh (56.4 ± 4.7%). In addition, we found that there was no difference between responses in two contexts in RG48-96/NaB and RG48-96/Veh at T3.

In the final stage, we decided to analyze whether the context memory impaired under DNMT inhibition can be facilitated under NaB administration if NaB was applied in 48 h after memory impairment and LTM was retested after that in 24 h. Three groups of snails were conditioned: RG72-96/NaB (*n* = 10), RG72-96/Veh (*n* = 9), and Veh/Veh (*n* = 8). Figure 4I–L shows that all groups acquired context memory (Figure 4J–L, T1, a reliable main effect of context F(1,48) = 152.5, *p* < 0.001). RG108 injections 24 h after test T1 resulted in a complete memory loss in T2 in groups RG72-96/NaB (6.5 ± 1.5%) and RG72-96/Veh (7.2 ± 1.5%), whereas there was no amnestic effect in the control group Veh/Veh (57.6 ± 7.0%). ANOVA revealed a reliable main effect of context F(1,48) = 224.9, *p* < 0.001; group F(2,48) = 7.0, *p* < 0.005; and a reliable interaction F(2,48) = 7.6, *p* < 0.005 (Figure 4J–L, T2). To determine whether NaB affected memory 48 h after RG108 administration, the RG72-96/NaB group received injections of NaB immediately after testing session T2; RG72-96/Veh and Veh/Veh were vehicle-injected at the same time point. A total of 24 h after NaB injection (Figure 4J–L, T3) ANOVA revealed a reliable main effect of group F(2,48) = 106.3, *p* < 0.001; context F(1,48) = 113.0, *p* < 0.001; and a reliable interaction F(2,48) = 95.0, *p* < 0.001. We found that neither vehicle nor NaB injections restored the conditioned response amplitude in RG108-treated animals (RG72-96/NaB—5.9 ± 0.8%, RG72-96/Veh—4.1 ± 1.0%). Post hoc analysis revealed that withdrawal responses on the ball in groups RG72-96/NaB and RG72-96/Veh at test T3 were significantly different from those in Veh/Veh (62.3 ± 8.9%). In short, NaB is not able to facilitate RG108-impaired context memory if more than 48 h had passed since the RG108 administration.

Thus, the results of behavioral tests clearly demonstrate that memory deterioration caused by the DNMT inhibitor RG108 could be rescued only if memory was reactivated in the first 48 h after administration of the DNMT inhibitor. After that time period the memory decline was irreversible. NaB, known as a cognitive enhancer, was able to reverse the effects of DNMT inhibition only within a narrow time window (should be less than 48 h after memory disruption).

## 3. Discussion

### 3.1. DNMT Inhibition Impairs Context Memory Maintenance but Not Reconsolidation

In the first section of the study, we focused on the role of DNA methylation as an epigenetic regulatory mechanism in the storage of the snail’s long-term context memory. In several series of behavioral experiments, we compared the effects of the DNMT inhibitor RG108 on LTM under various experimental conditions: RG108 was administered with or without reminding to determine dependence on the memory reactivation factor; testing was performed at various time intervals after RG108 administration to determine the temporal dynamics of the memory impairment. We did not detect any alterations in the snails’ withdrawal reactions in a safe context (on a glass) under RG108 administration, verifying that the observed withdrawal reactions changes were context-specific and not due to an unspecific RG108 effects. We have shown that long-term context memory in snails was impaired when the activity of DNMT was inhibited but was restored in one day after injection (Figure 1A). It should be noted that memory reactivation had a rescue effect on context memory during DNMT inhibition as inhibition of DNMT activity applied with retrieval had no effect on long-term context memory maintenance (Figure 1B). In further experiments, it was shown that when the interval between test sessions T1–T3 without memory reactivation was extended, memory recovery did not occur after RG108 administration (Figure 2A–C). Thus, the present study demonstrated that persistent DNA methylation plays a crucial role in memory maintenance in *Helix*. The fact that RG108 injection blocked expression of normal memory suggests that memory maintenance depends on the DNMT activity. According to the literature, the DNMT activity led to silencing of the memory-suppressor gene CREB2 in serotonin-induced plasticity in the marine mollusk Aplysia [33]. We can speculate that, at least partly, our results could be also associated with genes silencing: DNA methylation may inhibit memory-suppressor genes to protect the memory trace. Our results are in agreement with most previous studies; however, there is some discrepancy. Our findings are consistent with those in mammals [18,32,56], as well as in invertebrates [36,37,57]. However, a number of studies demonstrated that DNMT inhibition did not affect memory maintenance [22,26,58]. According to data concerning the effect of DNA methylation inhibition, the memory deficit caused by the DNMT inhibition is irreversible [26,31,36] which is inconsistent with our data. Regarding the reconsolidation process and the role of DNMT in it, many studies showed that inhibition of DNMT activity following memory reactivation led to impairment of memory reconsolidation [22,26,31,36,58]. Our results are not consistent with this fact. It is possible that the observed differences are associated with specifics of model organisms, as well as learning models. Time dependence of RG108 effects may be also a reason for different results.

An intriguing idea regarding the role of DNA methylation in memory processes was put forward [33,59]. According to it, the DNA methylation regulates the stability of a subset of connected neurons, or engram (roughly corresponding to “memory trace”) [60,61,62], that stores specific forms of memory, and DNA methylation is likely to be essential for preserving these engrams. Based on the results obtained, we can speculate that there are two distinct stages after the DNMT inhibitor administration. During the first stage (no more than 48 h after the DNMT inhibition) “the primary/residual memory trace” [36,49,63,64] is not completely impaired, which is confirmed by the possibility of memory recovery following testing. Thus, the first stage is characterized by transient impairment of mechanisms involved in the normal recall of LTM and preservation of “the primary/residual memory trace”. It seems that the molecular processes caused by DNMT inhibition develop in time, and only at the second stage (later than 48 h) we observe “the primary/residual memory trace” disruption and impossibility to activate the neural circuit underlying the long-term memory. At this stage the DNA methylation status is irreversibly violated, inter-connectivity of engram cells is lost, and, as a consequence, the reactivation of assemblies of connected engram neurons and effective memory recall are not possible.

### 3.2. The Rescue Effect of Context Memory Reactivation Depends on Nitric Oxide Synthesis, Protein Synthesis, and the Activity of the Serotonergic System

In the second section of the study, we focused on the mechanisms through which the long-term context memory impaired by DNMT inhibition could be restored. We explored how the process of memory recovery will be affected by the disruption of several biochemical mechanisms whose role in memory reconsolidation in *Helix* has been described earlier: NO synthesis [65], protein synthesis [66], and activation of the serotonergic system [67]. When snails were injected either with the NO synthase blocker L-NAME (Figure 3D), the protein synthesis blocker ANI (Figure 3E), or the serotonin receptor antagonist MET (Figure 3B) the memory recovery did not occur. Thus, the results showed that successful memory recovery depends on the simultaneous coordinated work of several systems: NO synthesis, protein synthesis, and the activity of the serotonergic system. Suppression of at least one of them with an appropriate blocker prevented memory recovery observed in control group (Figure 3C). These data confirm our assumption that the memory recovery process after the DNMT inhibition depends on an active reconsolidation process.

It is worth noting that there are several reports linking NO with changes in DNA methylation. According to them, the NO production upregulates DNMT activity and this effect could be fully prevented by NO synthase inhibitors [68,69,70,71]. Thus, the effect of L-NAME can be associated not only with its effect on the reconsolidation process but also with the effect on the activity of DNMT directly.

### 3.3. Inhibition of HDAC Activity Is Able to Rescue Context Memory Deficit Induced by a DNMT Inhibitor Only over a Limited Time Period

It can be expected that in the presence of different biochemical pathways of epigenetic regulation, in particular, DNA methylation and histone acetylation, the interaction of these mechanisms in memory processes is inevitable [5,20,26,55]. To investigate this issue in the final section of the study we explored whether the context memory impaired by DNMT inhibition could be restored by blockade of histone deacetylation. We set up three series of experiments in which DNMT were inhibited by RG108, followed by a reminder at different time intervals, followed by NaB administration. Similarly to the first experiment of this study (Figure 1A), we observed spontaneous memory recovery 48 h after RG108 injection (Figure 4A–D) if tested/reactivated each 24 h, and this recovery was significantly more pronounced under HDAC inhibition with NaB (compare Figure 4B,C) if tested in 48 h after RG108 injection. However, the memory was not restored if tested 72 h after RG108 administration, regardless of the presence of NaB injected at different time-points (Figure 4E–L). Thus, the memory-improving effect of both reminder alone and reminder in combination with HDAC inhibitor NaB was observed only over a narrow period of time (no more than 48 h after the RG108 administration). These results confirm our idea that during no more than 48 h after the DNMT inhibition “the primary/residual memory trace” [36,49,63,64] is kept but not expressed, and reactivation of memory triggers reconsolidation and restoration of memory which can be facilitated with HDAC inhibitor NaB.

The results in other studies [5,20,26,55] demonstrated that DNMT inhibition may control long-term plastic changes not only directly via changes in DNA methylation but also by influencing a histone acetylation. In these series of experiments pretreatment with a HDAC inhibitor rescued the memory impaired by a DNMT inhibitor. Thus, we can suggest another possible way via which the DNMT inhibition can regulate memory maintenance (the first one is silencing of memory-suppressor genes, see above) at least in part by preventing increase of histone acetylation, because treatment with NaB following reminding enhances the “rescue” effect of reminding itself.

## 4. Materials and Methods

Adult *Helix lucorum taurica* L. were used in these experiments. All animals had similar weights (15 g ± 5). A few days before conditioning animals were kept without food (to keep them active). The experimental procedures were in compliance with the Guide for the Care and Use of Laboratory Animals published by the National Institute of Health, and the protocol was approved by the Ethical Committee of the Institute of Higher Nervous Activity and Neurophysiology of RAS.

All behavioral procedures were conducted between 08:00 and 15:00 h. Animals were randomly allocated to experimental groups and tested in random order.

### 4.1. Behavioral Experiments: Context Training

The training procedure was similar to that in our previous studies of context memory in snails [72]. Animals were trained to distinguish between two contexts: experimental, in which they were shocked during training (ball, covered with aluminum foil), and control (flat glass). We estimated the defensive reaction of each snail (retraction of the posterior tentacles—ommatophores) in response to the same weak tactile stimulation of the rostral part of the leg in both contexts before and after training. The differences in withdrawal reactions in different contexts after training indicated context memory.

The first group of experiments was aimed at studying the role of DNMT activity in the maintenance of long-term context memory in snails and reconsolidation of this memory following retrieval (Figure 1).

The second group of experiments was aimed at studying whether the RG108-impaired memory by would be restored if memory was not reactivated in the first 24 h and 48 h after administration of the DNMT inhibitor (Figure 2).

In the third group of experiments, we investigated the mechanisms through which the mechanisms involved in reinstatement of DNMT inhibition-impaired context memory context memory can be reinstated (Figure 3). And, finally, we investigated whether the context memory impaired by the DNMT inhibition can be facilitated under the NaB (HDAC inhibitor) administration (Figure 4).

In all experimental groups, the test session T1 (test session after training) was taken as a starting point (0 h). Indices in group names indicate the time points from T1 when the tests T2 and T3 were made. The time when RG108 was injected was also counted from T1 (Table 1).

It should be noted that in our experiments we used the phenomenon of reactivation of a stored memory (reminder—R). In our case, it was placing an animal on a ball without shocks for 20 min (it is enough to cause memory reactivation and subsequent reconsolidation, as was shown in our previous studies [48]). The reminder leads to transient destabilization of consolidated memory and makes it labile and available for update.

### 4.2. Drugs and Injections

The protein synthesis inhibitor anisomycin (ANI, 0.4 mg for a snail), HDAC inhibitor sodium butirate (NaB, 4.8 µg/g of body weight), nonspecific blocker of serotonergic receptors methiothepin (MET, 5 µg/g of body weight); and DNMT inhibitor RG108 (all Sigma) were used in these experiments. ANI and RG108 were dissolved as described in [48] and [34] correspondingly. NaB and MET were dissolved in a sterile Ringer saline (in mM: 100 NaCl, 4 KCl, 7 CaCl2, 5 MgCl2, and 10 Tris-HCl buffer (pH 7.8)). Estimated final concentration in the hemolymph of free-behaving animals of ANI was 7.5 × 10^−5^ M, NaB—9 × 10^−6^ M, MET—1.4 × 10^−5^ M, and RG108—25 × 10^−6^ M. The injected volume of each drug was 0.1 mL. Control animals received the same volume of Ringer saline (vehicle). The concentrations applied were in accordance with the previous studies [34,48].

### 4.3. Data Analysis

Data are presented as mean ± SEM. Two-way ANOVA with one repeated measure (test) and post hoc Bonferroni test were used to evaluate the effects of drugs treatment. Significance was set at *p* < 0.05.

## 5. Conclusions

All things considered, our study demonstrated that DNMT inhibition regulates the maintenance of long-term context memory in *Helix*. Another important result is the reversibility of disrupting effect of DNMT inhibition on context memory maintenance. Our findings also demonstrated that DNMT inhibition did not affect the reconsolidation of context memory. We can only speculate regarding possible mechanisms concerning the role of DNA methylation in memory maintenance: the first is silencing of memory-suppressor genes; the second—preventing increase of histone acetylation. Further investigations that reveal downstream biochemical pathways caused by DNMT inhibition are needed. These findings also support the idea that DNA methylation is a common pathway for memory process regulation across different vertebrate and invertebrate species and behavioral paradigms.

## 6. Limitations of the Study

Our study have some limitations such as the lack of molecular evidence of the drugs’ effects that underlie the obtained results in the current study. Taking into account quite a few published studies in this field in all kinds of experimental animals, including gastropod snails, and demonstrating molecular evidence of protein synthesis blockade in neurons by anisomycine, inhibitory effects of RG108 on DNMT, blockade of NOS by L-NAME and NaB effects on HDAC, we have completely focused on behavioral research and carried out a large amount of experimental work, using all the necessary behavioral controls. Of course, the work would undoubtedly benefit from molecular controls in this animal, which we plan to obtain in future studies. This was not within the scope of the current study.

## Figures and Tables

**Figure 1 ijms-24-14068-f001:**
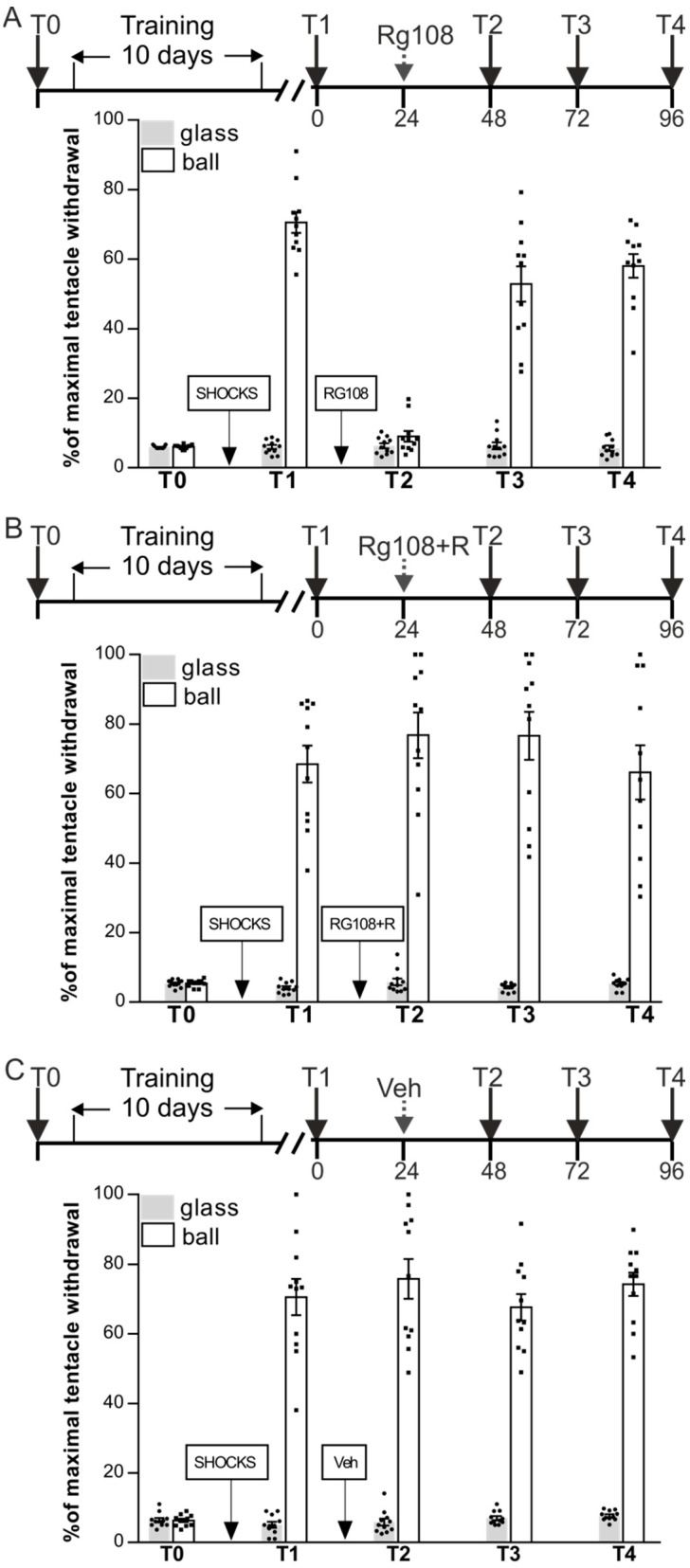
Inhibition of DNA methylation by RG108 injection affects the context memory maintenance. Group RG48-72 (*n* = 11) (**A**) was injected with RG108 without a reminder; group RG48-72 + R (*n* = 11) (**B**) was injected with RG108 with reminder (R—reminder, 20 min conditioned context re-exposure without shocks) next day after the test session T1; and group Veh (*n* = 11) (**C**) served as a control injected with vehicle. RG48-72 demonstrated no memory at T2 while RG48-72 + R and Veh groups exhibited good memory during test T2. A total of 24 h and 48 h later (T3 and T4), all groups (RG48-72, RG48-72 + R, and Veh) significantly discriminated against dangerous and safe contexts which means that memory in RG48-72 was reinstated.

**Figure 2 ijms-24-14068-f002:**
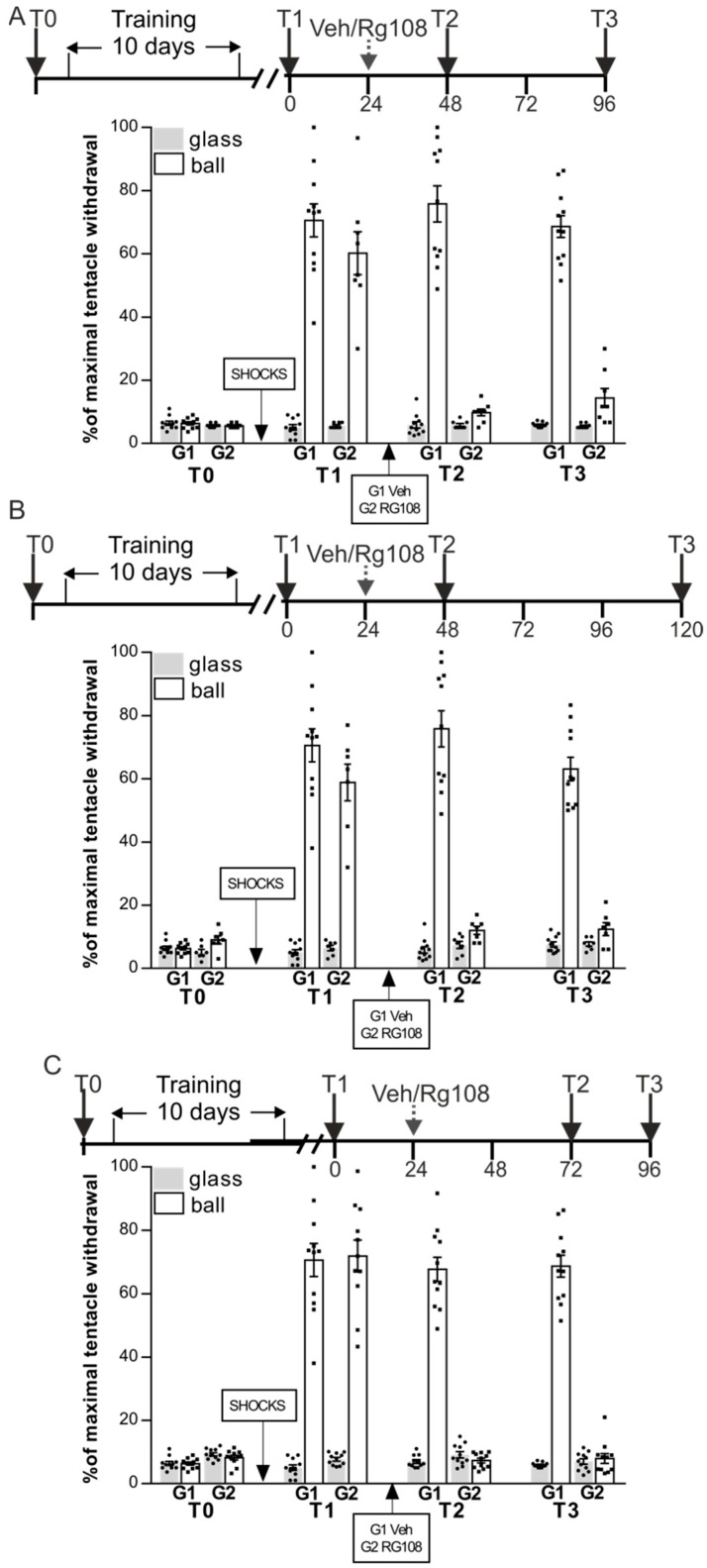
The memory-impairing effect of RG108 is irreversible without repeated memory reactivation: memory impaired by RG108 could not be restored if memory was not reactivated in the first 24 h and/or 48 h after administration of the DNMT inhibitor. (**A**) Context memory was retested in 24 h (T2) and 72 h (T3) after RG108 treatment. Group G2 (RG48-96, *n* = 8) was injected with RG108 without reminder; control group G1 (Veh, *n* = 11) was injected with vehicle. (**B**) Context memory was retested in 24 h (T2) and 96 h (T3) after RG108 treatment. Group G2 (RG48-120, *n* = 7) was injected with RG108 without reminder; group G1 (Veh, *n* = 11) served as a control and was injected with vehicle. (**C**) Context memory was retested in 48 h (T2) and 72 h (T3) after RG108 treatment. Group G2 (RG72-96, *n* = 11) was injected with RG108 without reminder; group G1 (Veh, *n* = 11) served as a control and was injected with vehicle. In all experiments (**A**–**C**) loss of memory caused by RG108 treatment was irreversible.

**Figure 3 ijms-24-14068-f003:**
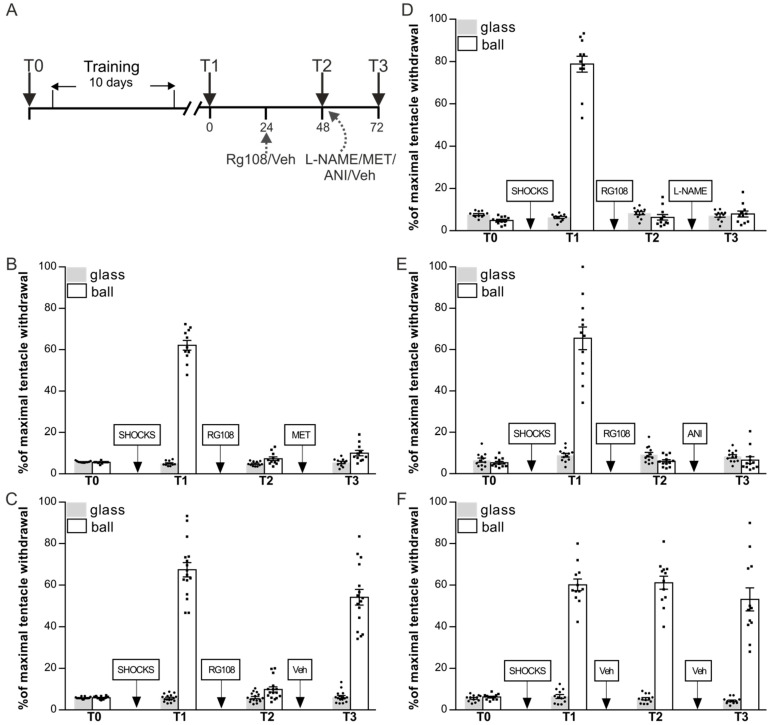
The effects of L-NAME, methiothepin (MET), and anisomycin (ANI) on the context fear memory reinstatement after DNMT inhibition. (**A**) Schematic representation of the experiment protocol. All groups RG48-72/MET (*n* = 11) (**B**), RG48-72/Veh (*n* = 16) (**C**), RG48-72/L-NAME (*n* = 13) (**D**) RG48-72/ANI (*n* = 12) (**E**), and Veh/Veh (*n* = 12) (**F**) were trained for 10 days in context conditioning and tested for retention 24 h later: all groups significantly distinguished dangerous (ball) and safe (flat glass) contexts (T1). (**B**) Methiothepin treatment. Inhibitor of DNA methylation RG108 was infused into snails 24 h later test session T1. Next-day test (T2) revealed that withdrawal responses were significantly reduced compared to that at test T1. Methiothepin was infused immediately after test session T2. When retested for retention 24 h later (48 h after RG108 injection), snails showed significantly lower withdrawal reactions compared to vehicle-treated controls Veh/Veh (**F**). (**C**) RG108 without memory reactivation significantly lowered withdrawal responses (test T2). Test session T2 (memory reactivation) had a rescue effect on context memory. Next-day test (T3) (48 h after RG108 injection) showed that impaired memory in group RG48-72/Veh was reinstated. (**D**) L-NAME treatment. RG108 administration produced amnestic effect at T2. Animals injected with L-NAME immediately after T2 showed no memory reinstatement at T3. (**E**) Anisomycin treatment. RG108 administration produced amnestic effect at T2. Animals injected with ANI immediately after T2 showed no memory reinstatement at T3. (**F**) Control group Veh/Veh injected at all stages with Ringer saline demonstrated good memory throughout all experiments.

**Figure 4 ijms-24-14068-f004:**
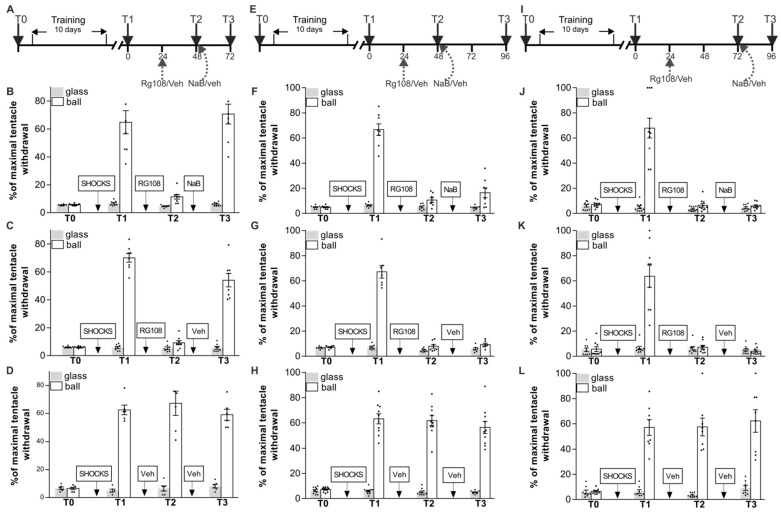
Effect of a single sodium butyrate (NaB) injection on context fear memory reinstatement after DNMT inhibition with RG108. (**A**–**D**) Schematic representation of the experiment protocol. All groups RG48-72/NaB (*n* = 8) (**B**), RG48-72/Veh (*n* = 8) (**C**), and Veh/Veh (*n* = 6) (**D**) were trained for 10 days in context conditioning and tested for retention 24 h later: they showed significant difference in responses between contexts (T1). The groups injected with RG108 24 h after T1, RG48-72/NaB and RG48-72/Veh, exhibited memory impairment during T2 in comparison with vehicle-injected control group Veh/Veh. Immediately after T2, snails were given injection of either NaB (RG48-72/NaB) or vehicle (RG48-72/Veh, Veh/Veh). The next day test (T3) (48 h after RG108 injection) showed that there was a significant difference in responses between contexts not only in Veh/Veh group but also between the RG-treated groups RG48-72/NaB and RG48-72/Veh, indicating that there was a positive effect of NaB on memory. (**E**–**H**) Schematic representation of the experiment protocol. All groups, RG48-96/NaB (*n* = 8) (**F**), RG48-96/Veh (*n* = 7) (**G**), and Veh/Veh (*n* = 10) (**H**), were trained for 10 days in context conditioning and tested for retention 24 h later: they showed significant difference in responses between contexts (T1). The groups injected with RG108 24 h after T1, RG48-96/NaB, RG48-96/Veh, exhibited memory impairment during T2 in comparison with vehicle-injected control group Veh/Veh. Immediately after T2 snails received injection of either NaB (RG48-96/NaB) or vehicle (RG48-96/Veh, Veh/Veh). Test 48 h later (T3) (72 h after RG108 injection) showed that there was a significant difference in responses between contexts only in the Veh/Veh group. There was no increase in withdrawal responses amplitudes in RG48-96/NaB and RG48-96/Veh, indicating that NaB administration did not affect memory impaired with the DNMT inhibitor when testing was delayed from the moment of RG108 administration by 72 h. (**I**–**L**) Schematic representation of the experiment protocol. All groups, RG72-96/NaB (*n* = 10) (**J**), RG72-96/Veh (*n* = 9) (**K**), and Veh/Veh (*n* = 8) (**L**), were trained for 10 days in context conditioning and tested for retention 24 h later: they showed significant difference in responses between contexts (T1). The groups injected with RG108 24 h after T1, RG72-96/NaB, RG72-96/Veh, exhibited memory impairment during T2 (48 h after RG108 injection) in comparison with vehicle-injected control group Veh/Veh. Immediately after T2 snails received injection of either NaB (RG72-96/NaB) or vehicle (RG72-96/Veh, Veh/Veh). Tests 24 h later (T3) (72 h after RG108 injection) showed that there was a significant difference in responses between contexts only in the Veh/Veh group. There was no increase in withdrawal responses amplitudes on the ball in both RG72-96/NaB and RG72-96/Veh groups, indicating that the memory was constantly impaired.

**Table 1 ijms-24-14068-t001:** Description of designations of experimental groups.

Group	Time Points When RG108 Was Injected and Tests T2 and T3 Was Performed
RG48-72	RG108 was injected 24 h after T1; T2 was performed 48 h after T1; and T3—72 h after T1
RG48-96	RG108 was injected 24 h after T1; T2 was performed 48 h after T1; and T3—96 h after T1
RG48-120	RG108 was injected 24 h after T1; T2 was performed 48 h after T1; and T3—120 h after T1
RG72-96	RG108 was injected 24 h after T1; T2 was performed 72 h after T1; and T3—96 h after T1

## Data Availability

Data will be provided available upon reasonable request.

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
