# Peer review of "DNA Methylation Inhibition Reversibly Impairs the Long-Term Context Memory Maintenance in Helix"

_ijms, 2023, doi:10.3390/ijms241814068_

Round 1
Reviewer 1 Report
This paper is interesting, only a few important points should be addressed:
- it is not clear what is the purpose of this paper. Please improve introduction section.
- "The fact that RG108 injection blocked expression of normal memory suggests that memory maintenance depends on... Aplysia" Look at and consider some intresting references: --PMID: 36010630 -- DOI: 10.3390/cells11162552 -- PMID: 31804189 -- DOI: 10.18632/aging.102558 -- PMID: 28161254 -- DOI: 10.1016/j.ijdevneu.2017.01.013
- Add a "limitations of the study" section, if any
- "Thus, the first stage is characterized by violation of mechanisms required for the normal expression.. engram. " Revise the sense of these sentences
- "These findings also support the idea that DNA methylation is a common pathway for memory... paradigms." What do authors mean?
Minor editing of English language required
Author Response
Please see the attachment.
We are very thankful to the Reviewers for detailed comments and analysis of our results.
Responses are marked in Yellow
A. Zuzina, A. Vinarskaya, P. Balaban

Reviewer 2 Report
Reviewer Comments
The present research article on “DNA-methylation inhibition reversibly impairs the long-term context memory maintenance in Helix”, has provided evidence demonstrating that an inhibitor of DNA methyltransferase (DNMT), RG108, impaired the long-term context memory in snails, and this impairment can be reversed in a limited time window, no more than 48 hours.
The research work is well-planned and the outcome of the study is presented nicely however paper have some limitation as mentioned below.
Scientific comments
1. What are the basic criteria for the selection of a gastropod mollusk Helix animal model for this study? However, among all the rodent behavioral models proved to be of the highest informative value.
2. What is the future significance of this study?
3. The lack of molecular study is a major limitation of the paper as mentioned by the authors in the conclusion.
4. The study lacks the immunoblotting-based study for NO synthase and serotonin receptors after injecting their blocker in snails.
Minor corrections/suggestions
1. In reference 17, unbold font 13.
Author Response

(The authors gave the same response as above.)

Reviewer 3 Report
The authors provide interesting data about RG108 effects on impairment of long-term context memory in snails. Although I am not an expert on memory consolidation, I have three main concerns: 1) the effect of RG108 on DNA methylation was never tested in the manuscript; furthermore, did L- NAME or methiothepin affect downstream the efficacy of RG108? I do not understand the link between L- NAME and RG108. Are NOS enzymes or arginine metabolism regulated by RG108? The effect of the release of NO must be very transient and limited to the site of production of NO. Have the authors quantified NO? What is the mechanism by which methiothepin counteracts the effect of RG108? I am surprised at the use of anisomycin in vivo, as I consider this treatment to be very dirty, nonspecific, and toxic. Other approaches, such as quantifying protein neosynthesis in RG108 treated snails, would be preferable. 2) The series of experiments with NaB is maybe not so logical; inhibitors of HAT must be used to restore a repressive chromatin environment. I would suggest removing this section or testing a HAT inhibitor. 3) How can the authors exclude new learning in the RG+R series of experiments?intermediate quality
Author Response

(The authors gave the same response as above.)

Round 2
Reviewer 3 Report
The study is of interest, however the authors have provided justifications based on their or others' prior scientific experience for the lack of controls I requested. Aware of this limitation, I ask the authors to introduce a paragraph explaining the limitations deriving from the lack of controls of their treatments.
Acceptable
Author Response
We have added in the text a rewritten section:
Limitations of the study
Our study have some limitations such as the lack of molecular evidence of the drugs’ effects that underlie the obtained results in the current study. Taking into account quite a few published studies in this field in all kinds of experimental animals, including gastropod snails, and demonstrating molecular evidence of protein synthesis blockade in neurons by anisomycine, inhibitory effects of RG108 on DNMT, blockade of NOS by L-NAME and NaB effects on HDAC, we have completely focused on behavioral research, and carried out a large amount of experimental work, using all the necessary behavioral controls. Of course, the work would undoubtedly benefit from molecular controls in this animal, which we plan to obtain in future studies. This was not within the scope of the current study.
(not in text) Just for estimation of the work that was done, we have to mention that it took about two years to collect and quantify all necessary series of behavioral experiments for this paper.
